# The Emerging Role of Autophagy as a Target of Environmental Pollutants: An Update on Mechanisms

**DOI:** 10.3390/toxics11020135

**Published:** 2023-01-30

**Authors:** Md. Ataur Rahman, Md Saidur Rahman, Md. Anowar Khasru Parvez, Bonglee Kim

**Affiliations:** 1Department of Pathology, College of Korean Medicine, Kyung Hee University, 1-5 Hoegidong Dongdaemun-gu, Seoul 02447, Republic of Korea; 2Korean Medicine-Based Drug Repositioning Cancer Research Center, College of Korean Medicine, Kyung Hee University, Seoul 02447, Republic of Korea; 3Department of Animal Science & Technology and BET Research Institute, Chung-Ang University, Anseong 17546, Republic of Korea; 4Department of Microbiology, Jahangirnagar University, Savar, Dhaka 1342, Bangladesh

**Keywords:** environmental exposure, autophagy, toxic materials, autophagosome, pesticides, particulate matter, nanoparticles

## Abstract

Autophagy is an evolutionarily conserved cellular system crucial for cellular homeostasis that protects cells from a broad range of internal and extracellular stresses. Autophagy decreases metabolic load and toxicity by removing damaged cellular components. Environmental contaminants, particularly industrial substances, can influence autophagic flux by enhancing it as a protective response, preventing it, or converting its protective function into a pro-cell death mechanism. Environmental toxic materials are also notorious for their tendency to bioaccumulate and induce pathophysiological vulnerability. Many environmental pollutants have been found to influence stress which increases autophagy. Increasing autophagy was recently shown to improve stress resistance and reduce genetic damage. Moreover, suppressing autophagy or depleting its resources either increases or decreases toxicity, depending on the circumstances. The essential process of selective autophagy is utilized by mammalian cells in order to eliminate particulate matter, nanoparticles, toxic metals, and smoke exposure without inflicting damage on cytosolic components. Moreover, cigarette smoke and aging are the chief causes of chronic obstructive pulmonary disease (COPD)-emphysema; however, the disease’s molecular mechanism is poorly known. Therefore, understanding the impacts of environmental exposure via autophagy offers new approaches for risk assessment, protection, and preventative actions which will counter the harmful effects of environmental contaminants on human and animal health.

## 1. Introduction

It has recently been shown that the cellular autophagy process, which involves lysosomes fusing with undesired or accumulated defective cellular components, is crucial for maintaining cellular function and homeostasis [1]. Autophagy is an active component of cell defense and helps cancer cells maintain their cytostatic link during the growth process. [2]. Phagophore assembly sites (PAS), which are structures that come before autophagosomes, are initiated by autophagy process. [3]. The endoplasmic reticulum (ER)-associated phosphatidylinositol 3-kinase (PI3K) is crucial for beginning the production of PAS [4]. AMP-activated protein kinase (AMPK), the mammalian target of rapamycin (mTOR), and unc-51-like autophagy activating kinase-1 (ULK1) facilitate phagophore formation during autophagy induction [5]. Phagophore recruiters include PI3K catalytic subunit type 3 (PIK3C3/Vps34), PI3K regulatory subunit 4 (PIK3R4/Vps15/p150), and beclin-1 (BECN1). Then, the membrane is expanded and sealed to lengthen it in preparation for autophagosome production. When autophagosomes reach maturity, they bind to lysosomes to create autolysosomes [6]. After acid hydrolases break down autolysosomes and their inner cargos, creating nutrients, additional metabolite recycling maintains cellular equilibrium [7] (Figure 1). The mTOR-independent autophagy mechanism has several therapeutic targets for neurodegenerative disorders [8]. mTOR-independent network regulating mammalian autophagy, encompassing cAMP-Epac-PLCϵ-IP3 and Ca2^+^-calpain-Gαs pathways, provides multiple therapeutic targets for neurological disorders. Enhancing autophagy through this mTOR-independent route is protective in different models [9].

Currently, global pollution threatens human health. Modern cultures worry about air pollution, including particulate particles and heavy metals. Due to their toxicity, endurance, and bioaccumulation, heavy metals, including cadmium, lead, and arsenic, are widespread contaminants [10]. Autophagy can be disrupted by various environmental pollutants, including pesticides, particulate particles, and heavy metals [11,12]. A single drug can have varying effects on the autophagy process depending on factors such as cell type, exposure length, and dosage. Therefore, understanding the effects of exposure to chemicals on autophagy has gained importance [13]. This understanding provides new avenues for environmental pollutant risk assessment, protection, and preventative measures that can be taken to protect against their adverse effects on human health [14].

Autophagy is a preprogrammed system that cells use to manage their internal homeostasis and reduce or eliminate the effects of foreign toxic chemicals that enter them (Figure 2) [15,16]. Therefore, autophagy induction may be a new restorative strategy in the toxicity field [17]. This review focuses on the autophagic pathways induced by various environmental metal pollutants to fill the knowledge gap on autophagy’s function in eliminating environmental toxicities.

## 2. Mechanism of Autophagy Pathways 

Numerous studies have been developed to reflect the different targeted subcellular component types that could be destroyed by autophagy [18]. Several autophagy receptors play a role in the autophagy pathways that are presented in Figure 3, including aggrephagy, mitophagy, nanoparticulophagy, reticulophagy, xenophagy, zymophagy, lipophagy, ribophagy, and pexophagy [19,20]. However, autophagy and its participation in the responses to various common environmental exposures such as metals, airborne particulate matter, nanoparticles, and cigarette smoke as well as some common single environmental toxins might be removed from the environment.

Autophagy selectively degrades lipids, called lipophagy, and sequestosome 1 (SQSTM1) autophagy receptors connect with lipid droplets through autophagy [21]. Mitophagy is the autophagic process that selectively degrades mitochondria [22]. During mitophagy, autophagy adaptors and receptors, such as the neighbor of BRCA1 gene 1 (NBR1), optineurin (OPTN), the nuclear dot protein 52 kDa (NDP52), and SQSTM1, recognize mitochondria [22]. During reticulophagy, the ER recognizes autophagy adaptors autophagy-related 40 (ATG40) and the ER-anchored autophagy receptor (reticulophagy regulator 1) [23]. In ribophagy, autophagy selectively degrades ribosomes, which bind to nuclear fragile X mental retardation-interacting protein 1 (NUFIP1), an autophagy receptor-like protein [24]. Midbody degradation is the selective destruction of midbody rings produced during cytokinesis via autophagy [25]. Midbody rings recognize autophagy receptors such as SQSTM1 and NBR1 during selective autophagy [26]. Pexophagy is the selective destruction of peroxisomes via autophagy; peroxisomes recognize autophagy adaptors autophagy-related 36 (ATG36), ATG40, and peroxisomal biogenesis factor 3 (PEX3), and autophagy receptors SQSTM1 and NBR1 [27]. Zymophagy selectively degrades damaged or surplus zymogen granules via autophagy; zymogen particles bind to autophagy receptors, including SQSTM1 [28]. Several proteins participate in selective autophagy, such as TAR DNA-binding protein 43 (TDP43), glucocerebrosidase (GBA), presenilin 1 (PSEN1), ATPase cation transporter 13A2 (ATP13A2), and superoxide dismutase-1 (SOD1) [29]. Selective autophagy processes contain proteins and specialized autophagic receptors which identify the cargo, generally mediated via cargo ubiquitination [30]. Through its interaction with a scaffold protein, the receptor either binds to cargoes or may be an integral component of these cargoes, connecting them to the autophagy machinery in the cell [31]. Protein aggregates are associated with Alzheimer’s, Parkinson’s, and Huntington’s diseases [2,32]. In yeasts, flies, and mammalians, cells mediate amyloid beta peptide, tau, poly-Q, alpha-synuclein, and mutant huntingtin protein aggregates which might be removed by aggrephagy, a selective disposal of protein aggregates [2,33]. Therefore, understanding the impacts of environmental pollutants on autophagy offers new ways for risk assessment, protection, and preventive actions to offset the harmful effects of environmental contaminants on human health.

## 3. Effects of Pesticides and Other Small Molecular Weight Environmental Toxins on Autophagy

Understanding the impacts of pesticides, small molecular weight, and chemical exposure on autophagy offers new ways for risk assessment, protection, and preventative actions to offset the harmful effects of environmental contaminants on human health [13]. Cellular components and protein kinases can activate several signaling pathways that result in autophagy, apoptosis, and necrosis [34]. Nanoparticles and metals act as strong autophagy activators in cell and animal systems [35]. When cells respond to metals/metalloids and nanoparticles, AMPK, mitogen-activated protein kinase (MAPK), AKT serine/threonine kinase 1 (AKT1/AKT), PI3K, death protein kinases, and mTOR are the main factors inducing or inhibiting autophagy [36]. Several receptors (e.g., NBR1, p62, Tax1 binding protein 1 [TAX1BP1], and OPTN) that recognize the autophagy adaptor (lipidated microtubule-associated proteins 1A/1B light chain 3A [MAP1LC3A/LC3; LC3II]) attract tagged mitochondria and facilitate autophagy vacuole engulfment [37]. Autophagy prevents chlorpyrifos (CPF)-induced reactive oxygen species (ROS)-mediated toxicity. CPF increases mitochondria-mediated apoptosis-related ROS production and autophagy in human neuroblastoma cells [38]. ROS connect environmental (pesticides, herbicides, heavy metals) and endogenous and genetic PD risk factors. Environmental toxins and medicines, such as 1-methyl-4-phenylpyridinium (MPP^+^), rotenone, paraquat (PQ), and metamphetamine, have been linked to autophagy dysregulation in neurotoxin-induced dopaminergic cell death models [39]. Epidemiological studies relate rural life, farming, well water, and agrichemicals to an increased risk of PD. Several agrichemicals harm dopaminergic neurons, suggesting an environmental foundation for sporadic PD [40]. Dopaminergic neurons are uniquely sensitive to the herbicide paraquat, with other populations of neurons unaffected, associated with diminished motor activity and dose-dependent striatal dopaminergic nerve fiber losses. Paraquat-treated animals showed upregulation and aggregation of -synuclein [-Syn] in the substantia nigra [41]. Anti-apoptotic proteins interact with BECN1 and BCL2-associated X apoptosis regulator (BAX) or BCL2 antagonist/killer 1 (BAK1/BAK) [42,43]. Interestingly, BECN1 is a PI3K component and autophagy activator that normally interacts with anti-apoptotic proteins (e.g., B-cell lymphoma 2 [BCL2]) to suppress autophagy [44]. Stress interrupts the connection, promoting autophagy. Stress affects the connection between BCL2 and BAX/BAK, increasing apoptosis [45]. During continued stress, BECN1 is cleaved by caspase and translocates to the mitochondria, increasing apoptosis [46]. Depending on its length, BECN1 can induce autophagy or apoptosis [43]. In stressed cells, calpain or caspases also degrade autophagy-related 5 (ATG5) and autophagy and beclin-1 regulator 1 (AMBRA1), shifting autophagy towards apoptosis [47]. Moreover, arsenic inhibits p62-mediated selective autophagy, stabilizing FTO protein. FTO overexpression can prevent autophagy, maintaining FTO accumulation in a positive feedback loop [34]. Physical, chemical, and biological processes that occur in plants, animals, and humans exposed to environmental toxins that result in autophagy are presented in Figure 4.

Autophagy and membrane trafficking degrade and recycle macromolecules in lysosomes. It has also been found that lysosomal membranes and ATP-dependent proton pumps keep the lumen acidic for enzyme activity [47]. However, lumen pH or lysosomal membrane permeability variations cause lysosome dysfunction, disrupting the autophagosome–lysosome fusion [48]. Changes in membrane permeability can produce acidification and necrosis. Perfluorooctanoic acid, arsenic, and cadmium inhibit lysosomal functioning [49]. Moreover, rotenone increases ROS production, inhibiting the mitochondrial electron chain and causing autophagy in neuroblastoma cells [50]. Longer exposure causes autophagosome accumulation and lysosomal pH disruption. Malathion causes autophagosome accumulation in SH-SY5Y cells [51]. It inhibits acetylcholinesterase, causing neurotoxicity, and destabilizes lysosomal membranes, impairing autophagosome–lysosome fusion and causing autophagosome buildup. 

Agriculture uses the neurotoxic pesticide fipronil [52]. Pre-treating SH-SY5Y cells with rapamycin enhanced cell viability after fipronil treatments and reduced apoptosis [13]. N-acetylcysteine, a ROS scavenger, reduced fipronil-induced autophagy and apoptosis, showing that oxidative stress is required for toxicity and autophagy [13]. Polybrominated diphenyl ethers BDE-153 and BDE-100 induced autophagy in human liver HepG2 cells through oxidative stress and mitochondrial dysfunction (mitophagy) [53]. Recent research suggests that direct oxidation of catalytic thiol-groups on autophagy-related 3 (ATG3) and 7 (ATG7) might block LC3’s conjugation with phosphatidylethanolamine, which is essential for effective autophagy [54]. ER stress promotes ROS production and can affect redox equilibrium in the cell. Particularly, cadmium, perfluorooctanoic acid (PFOA), paraquat (PQ), cigarette smoke, and chloropicrin trigger ER stress and autophagy. ER stress is caused by inadequate protein folding or diminished ER folding capacity [55]. In particular, PQ is a non-selective herbicide that induces ER stress and autophagy in SH-SY5Y cells [56]. However, inhibiting autophagy increased apoptosis, suggesting it protects against PQ-induced toxicity. PFOA disrupts lipid metabolism, increases ROS levels, and causes ER stress. It induced autophagic vacuole accumulation and disrupted autophagosome–lysosome fusion in the mouse liver in vivo and in a human hepatocyte culture in vitro [57].

## 4. Targeting Autophagy Modulation to Eliminate Environmental Substances 

Mammalian cells use selective autophagy, a critical process, to destroy environmental toxins and damaged organelles without damaging cytosolic elements. Depending on the autophagy receptors and cargo targeted, selective autophagy can be categorized as either inducing or inhibiting.

### 4.1. Elimination of Particulate Matter by Autophagy

Air pollution has emerged as a significant problem in the environment, particularly due to the presence of extremely minute pollutant particles and pathogenic microbes, which can cause significant harm to the human body. Filtration of the air is one method of cleaning the air that has proven to be both popular and successful [58]. Particulate matter (PM) comprises tiny particles or liquid droplets that are so tiny that they can be inhaled and cause significant harm to an individual’s health [59]. PM, metals, black carbon, nitrate, organic aerosols, polycyclic aromatic hydrocarbons, automotive exhaust, and sulfates comprise microscopic particles and liquid droplets floating in the air [60]. PM has been associated with various health conditions, especially respiratory illnesses. Several studies have associated PM exposure with autophagy and airway dysfunction [61,62]. PM activated the nuclear factor kappa-light-chain-enhancer of the activated B cells (NF-κβ) pathway, airway inflammation, and mucus hyper-production in human bronchial epithelial (HBE) cells [63]. Fine PM triggered cytotoxicity and enhanced autophagy, oxidative stress, and the tumor necrosis factor (TNF/TNFα) pathway in human lung epithelial cells [64]. It has been reported to activate autophagy and inflammation in HBE cells in vitro and in vivo [65]. An autophagy inhibitor (3-methyladenine) suppressed PM-activated pro-inflammatory cytokine expression in vitro and in vivo in PM-treated mice [66]. Additionally, diesel exhaust particle exposure triggered autophagy and citrullination in normal HBE (NHBE) cells. Both Euro 4 and Euro 5 carbon particles could severely alter cell viability, inducing autophagy, apoptosis, and necrosis and stimulating pro-inflammatory cytokine interleukin (IL)-18 production, protein citrullination, and protein arginine deiminase activity in NHBE cells [67]. Therefore, PM has been associated with several health problems in humans, notably respiratory conditions, which can be reduced via the autophagy pathway.

### 4.2. Elimination of Nanoparticles by Autophagy

A nanoparticle, a type of ultrafine particle, is often described as a particle of matter with a diameter of 1-100 nm. Nanoparticles exist naturally and are studied in chemistry, physics, geology, and biology [68]. One of the mechanisms of intrinsic toxicity that are exhibited by NPs is the disruption of autophagy. The disruption of autophagy that NPs cause must be understood in order to ensure the safety of nanotechnology [10]. Autophagy induced by nanoparticles via endocytosis or other routes may have therapeutic effects, indicating biological applications, although their processing and destruction via selective autophagy remains unknown. Nanoparticles are foreign entities that are destroyed by cells [69]. Nanoparticles enter cells via endocytosis or other uptake routes. Nanoparticles colocalize with autophagy receptors or markers to produce nanoparticle-containing autophagosomes called nanoparticulosomes [70]. These ubiquitinated nanoparticles engage with autophagy receptor proteins, SQSTM1 bound to LC3, forming an autophagosome (Figure 5) [70]. Nanoparticles affect autophagy by increasing autophagosome production and flux or causing autophagic malfunction [71]. Nanoparticles enhance LC3 levels in various categories [35]. In autophagy malfunction, SQSTM1 levels increase because it is no longer degraded. Carbon nanotubes, poly(amidoamine) dendrimers, iron oxide nanoparticles, and graphene oxide caused autophagosome accumulation by blocking autophagic flow [72]. Silver and iron oxide nanoparticles induce autophagy by producing ROS [73]. Alumina, fullerenes, cationic dendrimers, carbon nanotubes, quantum dots, gold, zinc oxides, and silica were found to activate autophagy by blocking mTOR or promoting the expression and phosphorylation of autophagy-related proteins [74]. Lanthanum oxide, cerium dioxide, europium oxide, and manganese also triggered autophagy [75]. Caveolin 1 (CAV1) is an important membrane protein for cell membrane trafficking and autophagy [76]. Additionally, biodegradable ferric phosphate nanosheets are coated with doxorubicin for targeted tumor eradication via an autophagy inhibition-enhanced apoptosis/ferroptosis pathway [77]. Therefore, nanoparticulophagy shows intracellular trafficking mechanisms other than degradation routes for digesting nanoparticles and nanodrugs with therapeutic and pathophysiological consequences.

### 4.3. Elimination of Toxic Metals by Autophagy

Metals and metalloids are human and environmental toxicants. Recently, autophagy has been studied for eliminating physicochemical metal factors that exacerbate toxicity [78]. Autophagy begins when the flat membrane wraps around cytosol or organelles, forming the double-membrane autophagosome vesicle. During autophagosome formation, membranes expand and form a cup-like phagophore [79]. Generally, phagophores are formed by isolating the original membrane and assimilating lipids or repurposing existing compartments. Vesicles sequester cytosolic material and transport it to the lysosomal lumen, forming single membrane autophagolysosomes that digest their contents [80]. The lysosomal membrane-associated protein (LAMP) maintains cellular homeostasis through ubiquitination during maturation. Light chain 3B (LC3B) is ubiquitinated to form an integral membrane protein complex in a nascent autophagosome [8]. Autophagy’s molecular basis could be a technique for removing toxic, hazardous metals from the environment (Figure 6). Arsenic causes DNA damage, apoptosis, and oxidative stress [15]. Arsenic’s short-term activation of autophagy protects against apoptosis [81]. Prolonged exposure to environmental dosages impairs autophagy. Cadmium promotes DNA strand breaks, ER stress, ROS production, and calcium homeostasis [82]. Prolonged contact with these metals reduces the rate at which the p62 protein is degraded by autophagy, resulting in its accumulation [83]. BCL2 interacting protein 3 (BNIP3) is essential for arsenic trioxide (As_2_O_3_)-induced autophagy in malignant glioma cells [84]. As_2_O_3_-induced autophagic cell death involves LC3 and mitochondrial membrane rupture but not caspase activation [85]. As_2_O_3_ is a powerful autophagy inducer that appears to need MAPK kinase (MEK)/extracellular signal-regulated kinase (ERK) pathway activation but not MAPK8/JNK or AKT/mTOR [86]. However, arsenic induces autophagy, modifies autolysosomal gene expression, and inhibits cellular growth in human lymphoblastoid cell lines [87]. Additionally, the Ca^2+^-mitochondrial-caspase and Ca^2+^-ERK-LC3 signaling pathways increase cytosolic cadmium levels to promote autophagy and cell death in MES-13 cells [88]. Cadmium accumulated in rat kidney proximal convoluted tubule lysosomes, stimulating cell growth and autophagy [89]. However, further investigations on the roles of arsenic and cadmium in triggering autophagic cell death are needed.

In addition, mercury (Hg) toxicity causes DNA damage, suppresses DNA and RNA synthesis, and induces protein structural changes in vivo and in vitro [90]. Hg poisoning triggers autophagy in rat hepatocytes by modulating the ATG5-autophagy-related 12 (ATG12)-LC3B covalent-conjugation pathway via ubiquitination [11]. In response to Hg, autophagy monitors cell fate by recruiting caspase-8 (CASP8) to autophagosomes via its Fas-associated death domain [91]. High chromium (Cr[III]) damaged DNA [92]. Hexavalent Cr can trigger autophagy in stem/progenitor cells. Stem/progenitor cells subjected to subtoxic and toxic Cr concentrations had preserved tissue regeneration potential [93]. Autophagy indicates Cr toxicity in cord blood hematopoietic stem cells [94]. However, the hematopoietic lineage responds to Cr(VI)-mediated toxic stress via apoptosis and autophagy. Molecular switching between these two pathways may be mediated by stem/progenitor cell differentiation [95]. Iron (Fe) excess causes brain necrosis and apoptosis. DNA damage and oxidative stress exacerbate Fe^2+^-mediated toxicity [96]. Fe^2+^-mediated cell death is not necessarily via apoptosis. Recent studies showed that upregulating the ferritin stress protein complex is a quick adaptation mechanism, with ferritin autophagy influencing cellular susceptibility to the oxidative stress response [97]. Nuclear receptor coactivator 4 (NCOA4) acts with GABA type A receptor-associated protein-like 2 (GABARAPL2/ATG8) to recruit a specific cargo-receptor complex into autophagosomes, called ferritinophagy, which is crucial for Fe homeostasis [98]. Therefore, it has been suggested that mammalian cells use autophagy as a cytoprotective defense against several types of metabolic toxicity or organelle damage.

### 4.4. Elimination of Smoke by Autophagy

In the indoor environment, one of the most significant contributors of particulate matter and chemical pollutants is the act of smoking. A combination of the main stream of smoke that is expelled from the lungs of smokers and the side stream of smoke that is generated straight from the burning cigarette, pipe, or cigar is what makes up second-hand smoke [99]. Epoxide hydrolase 2 (Ephx2)-deficient animals were found to have less lung inflammation and autophagy due to cigarette smoke exposure than normal mice [100]. The autophagy signaling pathway is enhanced by nicotine exposure, causing the heart to adopt an ischemic-sensitive phenotype. It offers an autophagy suppression therapeutic approach that may be innovative for treating ischemic heart disease [101]. The immunological response is commonly associated with autophagy activation caused by cigarette smoke. In cigarette-exposed mice, the rise in pulmonary p62 is highly linked with increased expression of bicaudal D1 (BICD1), an adapter protein that binds to the dynein motor apparatus linking microtubule transport to lysosomes [102]. Recently, cigarette smoke was found to induce autophagy and trigger immune- and oxidation-related responses that damage the airway and alveolar epithelium [103]. The development of smoke aggregates, generally cleared by autophagy, is under the control of the multifunctional protein p62 [104]. However, in vitro exposure of bronchial epithelial cell line BEAS-2B to cigarette smoke extract causes ubiquitinated protein aggregates that colocalize with LC3B and p62 [105]. Carbamazepine reduces these aggregates. In mice exposed to cigarette smoke, aggresomes, LC3B, and p62 increase in peripheral lung tissue, correlating with cellular senescence [106]. Recently, cigarette smoke was shown to be responsible for the accretion of an additional autophagy-related protein called the transcription factor EB (TFEB) in the mouse lung in vivio and HBE cells in vitro [107]. In airway epithelial cells, mTOR was found to regulate cigarette smoke-activated apoptosis, autophagy, inflammation, and necroptosis [108,109]. In stable chronic obstructive pulmonary disease (COPD), the majority of studies have demonstrated an impairment in autophagy, with reduced autophagic flux and accumulation of abnormal mitochondria (defective mitophagy), and are linked to cellular senescence [110]. Acute exposure to cigarette smoke may activate autophagy, resulting in ciliary dysfunction and death of airway epithelial cells [111]. It is challenging to target autophagy therapeutically since the level of autophagy might vary from cell type to cell type and from one environment to another inside a cell [112]. However, these medications are not specific, and researchers are currently working on developing drugs that are more selective. These drugs have the potential to be beneficial as innovative agents in the treatment of asthma and COPD in the future.

## 5. Conclusions

Several recent studies have found that distinct signaling pathways ultimately resulting in autophagy are activated in cells by cellular proteins or kinases as a protective response to environmental toxins and are associated with increased cell survival. Many studies indicate that autophagy plays a key role in cellular reactions to environmental toxins. However, its significance in environmental toxicant exposures remains unknown. Therefore, environmental chemical exposures must be further studied to determine autophagy’s molecular pathways. However, unknown relationships exist between autophagy, immunological responses, and other cellular activities induced by chemical stress. Additionally, novel technologies and animal models will be needed to identify the complicated autophagic routes in metal-induced cytotoxicity. Therefore, further studies are urgently needed to explore how chemicals affect autophagy. These findings will be important for undertaking risk assessments, protective measures, and prevention activities for environmental contaminants with health impacts. Therefore, determining how to apply autophagy in environmental areas related to human health and establishing associations between autophagy and environmental exposure are appealing subjects for further research.

## Figures and Tables

**Figure 1 toxics-11-00135-f001:**
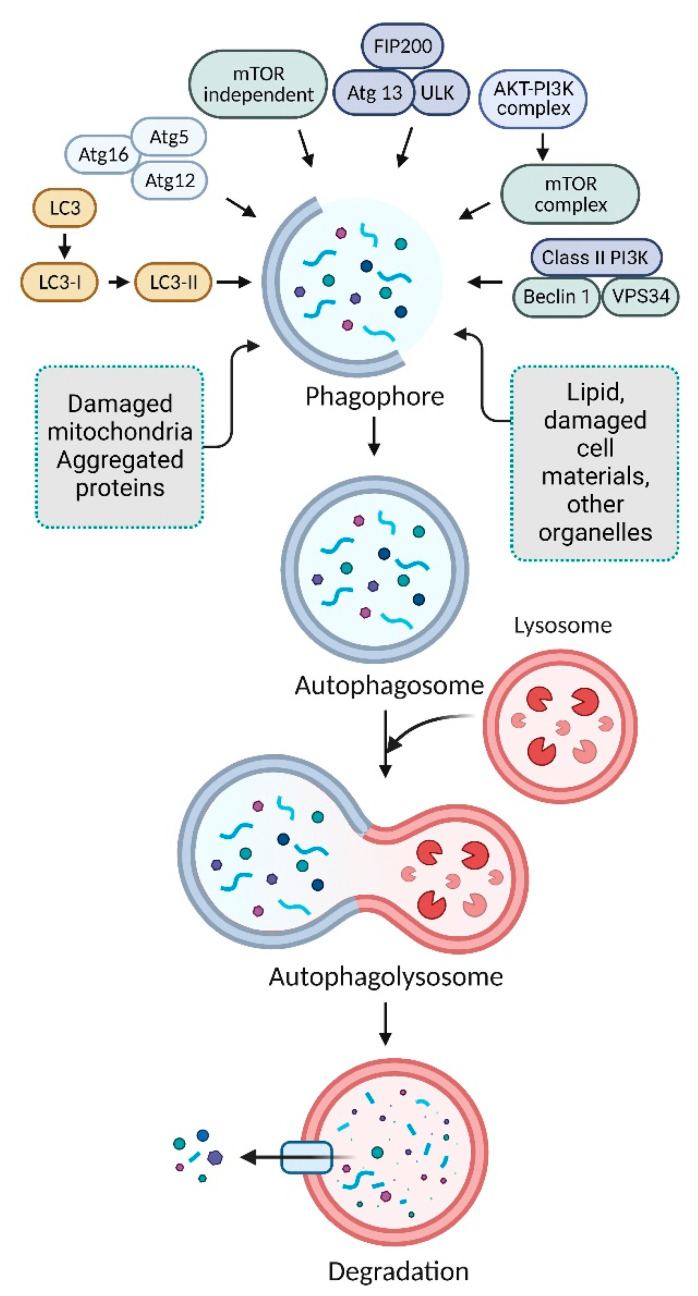
The autophagic pathway’s molecular mechanism. The development of a pre-autophagosomal structure triggers autophagy. The pre-autophagosomal structure is partly formed by PI3K-AMPK, and mTOR. The BECN1 complex, ULK1, Vps34, and phagophore production are all stimulated. Phagophore nucleation is extended, followed by autophagosome binding. Autolysosomes are created when a mature autophagosome binds to a lysosome. Acid hydrolases finally destroy autolysosomes, creating nutrients and recycling metabolites.

**Figure 2 toxics-11-00135-f002:**
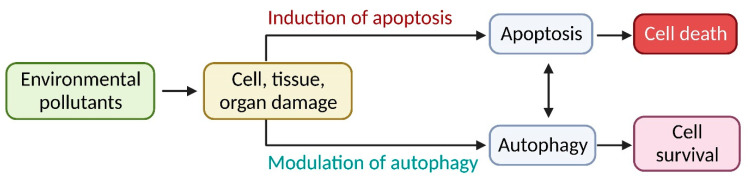
Interaction between the many distinct cell death types and survival induction by environmental pollutants.

**Figure 3 toxics-11-00135-f003:**
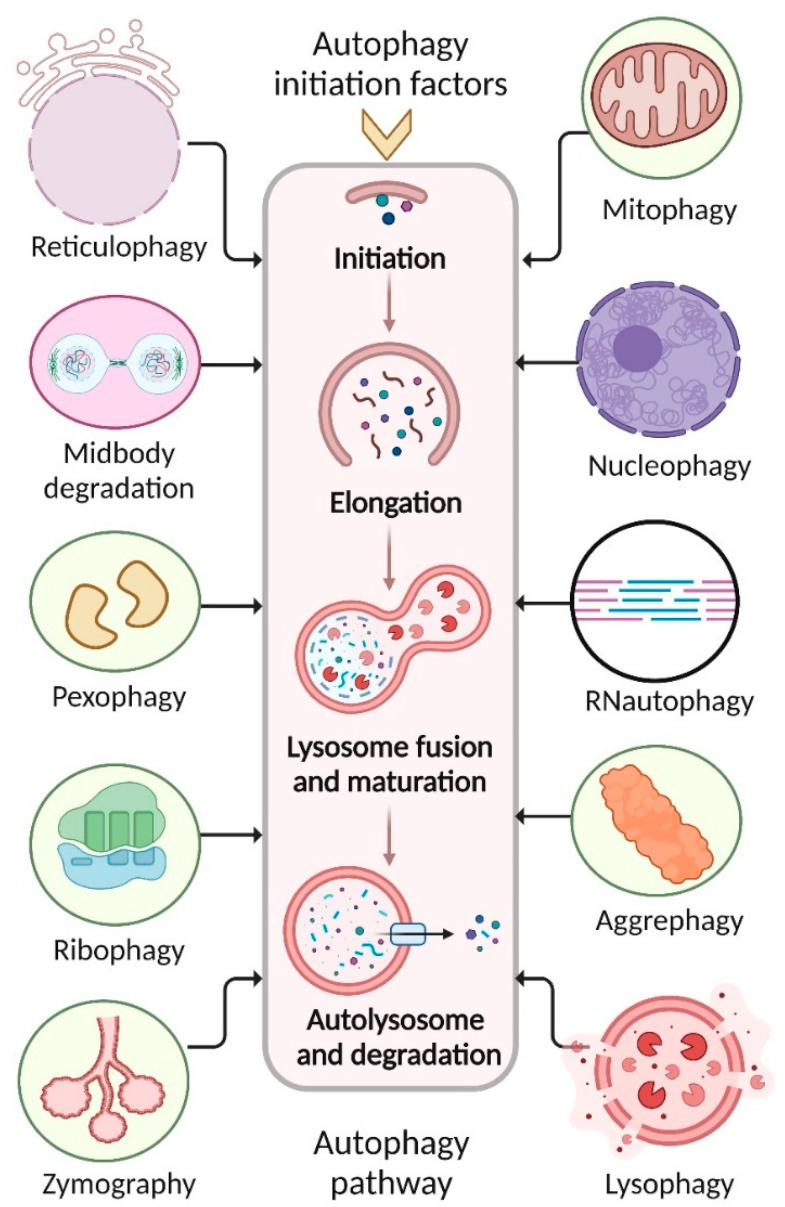
Many selective autophagy processes contribute to environmental toxin removal.

**Figure 4 toxics-11-00135-f004:**
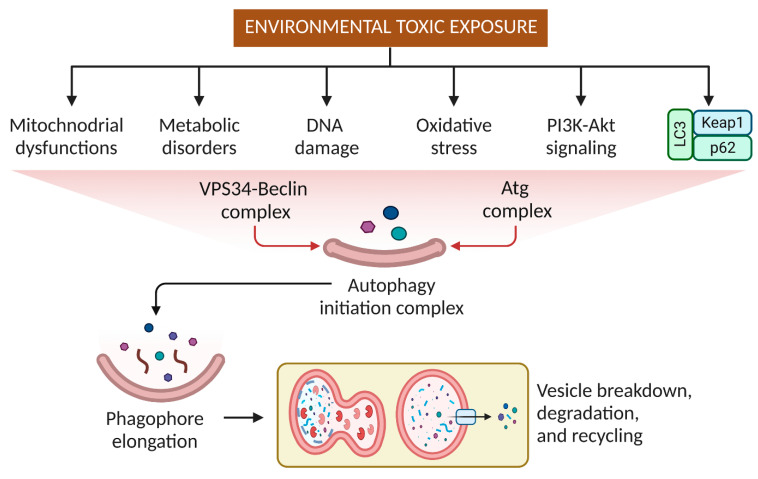
Physiological and biochemical reactions in plants, animals, and humans in response to environmental toxin exposure.

**Figure 5 toxics-11-00135-f005:**
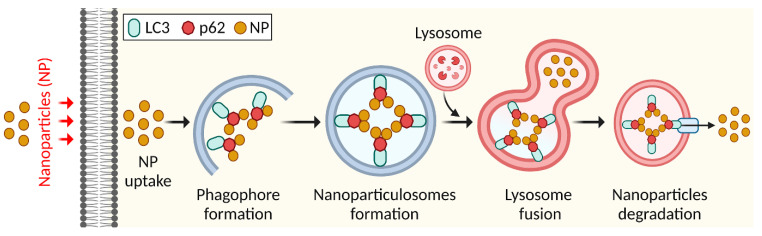
Schematic diagram of nanoparticles eliminated by autophagy.

**Figure 6 toxics-11-00135-f006:**
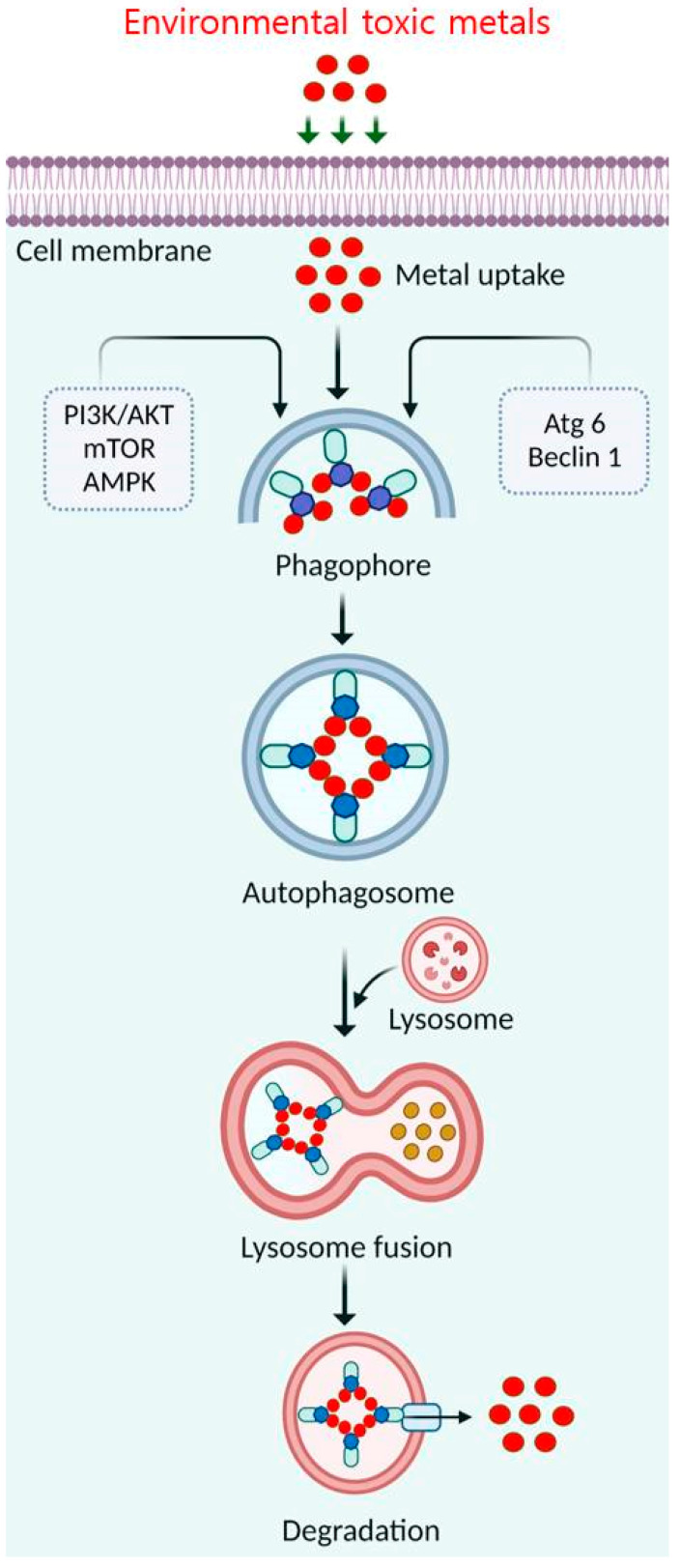
Schematic diagram and molecular mechanism of autophagy as a method for removing hazardous toxic metals from the environment.

## Data Availability

Request upon corresponding author.

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
