# Peer review of "The Emerging Role of Autophagy as a Target of Environmental Pollutants: An Update on Mechanisms"

_toxics, 2023, doi:10.3390/toxics11020135_

Round 1
Reviewer 1 Report
The manuscript entitled on"Recent Update and Emerging Role of Autophagy Modulation to Eliminate Environmental Vulnerability from molecular mechanisms and developments" by Md. Ataur Rahman et.al tried to review the molecular mechanism and development of autophagy regulation to eliminate environmental vulnerability. The research is interesting. Before the manuscript is suggested for publication in the journal, some concerns need to be addressed.
1. The pictures in the manuscript are good and clear.
2. Many abbreviations in the manuscript do not indicate their full name when they appear for the first time.
3. Various autophagy pathways found at present are reviewed in “Recently studies autophagy pathways to target environmental toxic clearance”, but the autophagy pathways of environmental toxicants and environmental toxic substances are not mentioned. The author should consider modifying the subtitle or adding environmental toxic substances.
4. In the manuscript “Recent Update and Emerging Role of Autophagy Modulation to Eliminate Environmental Vulnerability from molecular mechanisms and developments”, both the title and abstract mention the role of autophagy in the elimination of harmful substances in the environment. However, there are many other factors in the text that affect autophagy, such as stress, cigarettes, please point out the definition and scope of the environment.
5. From the perspective of the title, the author attempts to review the molecular mechanism and development of autophagy regulation to eliminate environmental vulnerability, but the paper does not mention the specific mechanism in detail, most of which are listed without more discussion.

Author Response
The manuscript entitled on" Recent Update and Emerging Role of Autophagy Modulation to Eliminate Environmental Vulnerability from molecular mechanisms and developments" by Md. Ataur Rahman et.al tried to review the molecular mechanism and development of autophagy regulation to eliminate environmental vulnerability. The research is interesting. Before the manuscript is suggested for publication in the journal, some concerns need to be addressed.
>> (Response) First of all, we would like to express our sincere gratitude for the time and effort the reviewer had put into reviewing our manuscript. We have incorporated changes based on the reviewer comments provided in the manuscript which revised parts are highlighted by BLUE color in the entire revised manuscript.
- The pictures in the manuscript are good and clear.
>> (Response) Thank you for reviewer for nice comments.
- Many abbreviations in the manuscript do not indicate their full name when they appear for the first time.
>> (Response) We checked modified all the abbreviations through the entire manuscripts.
- Various autophagy pathways found at present are reviewed in “Recently studies autophagy pathways to target environmental toxic clearance”, but the autophagy pathways of environmental toxicants and environmental toxic substances are not mentioned. The author should consider modifying the subtitle or adding environmental toxic substances.
>> (Response) We modified the subtitle by adding environmental toxic clearance. 2. Autophagy pathways to target environmental toxin substances clearance. Page 5, line 81.
- In the manuscript “Recent Update and Emerging Role of Autophagy Modulation to Eliminate Environmental Vulnerability from molecular mechanisms and developments”, both the title and abstract mention the role of autophagy in the elimination of harmful substances in the environment. However, there are many other factors in the text that affect autophagy, such as stress, cigarettes, please point out the definition and scope of the environment.
>> (Response) We added discuss the factor cigarettes in section 4.4 ‘Elimination of smoke by autophagy’. Page 13, line 285-290.
- From the perspective of the title, the author attempts to review the molecular mechanism and development of autophagy regulation to eliminate environmental vulnerability, but the paper does not mention the specific mechanism in detail, most of which are listed without more discussion.
>> (Response) We mentioned the specific mechanism of detail in the following’s points with figures. Due to the paper lengths we brief discuss the points.
4.1 Elimination of particulate matter by autophagy, page 9.
4.2 Elimination of nanoparticles by autophagy with figure 5, page 10.
4.3 Elimination of toxic metal by autophagy with figure 6, page 11.
4.4 Elimination of smoke by autophagy, page 13.

Reviewer 2 Report
The review by Rahman et al. describes different findings from recent literature on how environmental pollutants affect autophagy pathways. The review focuses on different types of solid dust particles, nanoparticles, metal and metalloids and smoke. The topic is really interesting and updated reviews are missing. The manuscript is accompanied by very nice illustrations and many very recent references. Nevertheless, the content of this review is difficult to follow and often incomprehensible. Starting from the title and finishing with the conclusions paragraph, the authors fail to deliver a clear message to the reader. In many cases it is a problem of bad English grammar, but very often the main reason seems to be a chaotic and poorly organized way of communicating scientific facts. These major points of criticism are listed below.
Major points:
1. The title is difficult to understand and fails to clearly deliver the main message of the review. It is also too long and English grammar is poor. Much better title could be: “Emerging role of autophagy as a target of environmental pollutants.”
2. The abstract is also too long and confusing as it does not specify clearly enough what will be discussed in this article. There are too many generic sentences and terms written in relatively comprehensible English, which say very little on the purpose and content of the review. The authors should shorten the abstract to 250 words and clearly state the purpose and main points that are discussed in the review.
3. The review is properly organized in chapters and paragraphs, but many subtitles are incomprehensible or poorly formulated. Just few examples:
- Line 87: Recently studies autophagy pathways to target environmental toxic clearance
- Line 123: Molecular target of autophagy for environmental toxic management
- Line 179. Recently target and application of autophagy to eliminate environmental exposure
4. The same is true for at least one third of sentences in the content of most paragraphs. The valuable scientific findings cannot be delivered properly with such an overwhelming linguistic problem. For the same reason it is pointless to list all badly formulated sentences. The authors should rewrite the review after seeking the help of any researcher who is proficient in English.
5. Paragraphs 4.1 and 4.2: the terms “nanoparticles” and particulate matter should be clearly defined.
Author Response
The review by Rahman et al. describes different findings from recent literature on how environmental pollutants affect autophagy pathways. The review focuses on different types of solid dust particles, nanoparticles, metal and metalloids and smoke. The topic is really interesting and updated reviews are missing. The manuscript is accompanied by very nice illustrations and many very recent references. Nevertheless, the content of this review is difficult to follow and often incomprehensible. Starting from the title and finishing with the conclusions paragraph, the authors fail to deliver a clear message to the reader. In many cases it is a problem of bad English grammar, but very often the main reason seems to be a chaotic and poorly organized way of communicating scientific facts. These major points of criticism are listed below.
>> (Response) First of all, we would like to express our sincere gratitude for the time and effort the reviewer had put into reviewing our manuscript. We have incorporated changes based on the reviewer comments provided in the manuscript which revised parts are highlighted by BLUE color in the entire revised manuscript. Furthermore, professional language editors reviewed and improved the quality of our manuscript's English (Company name: Cambridge Proofreading LLC, Invoice No: 239-35-15).
Major points:
- The title is difficult to understand and fails to clearly deliver the main message of the review. It is also too long and English grammar is poor. Much better title could be: “Emerging role of autophagy as a target of environmental pollutants.”
>> (Response) We modified the title accordingly ‘The emerging role of autophagy as an environmental pollutant target: a mechanism updates’
- The abstract is also too long and confusing as it does not specify clearly enough what will be discussed in this article. There are too many generic sentences and terms written in relatively comprehensible English, which say very little on the purpose and content of the review. The authors should shorten the abstract to 250 words and clearly state the purpose and main points that are discussed in the review.
>> (Response) We checked and modified shorten the abstract below 250.
- The review is properly organized in chapters and paragraphs, but many subtitles are incomprehensible or poorly formulated. Just few examples:
- Line 87: Recently studies autophagy pathways to target environmental toxic clearance
- Line 123: Molecular target of autophagy for environmental toxic management
- Line 179. Recently target and application of autophagy to eliminate environmental exposure
>> (Response) We checked and modified accordingly.
- The same is true for at least one third of sentences in the content of most paragraphs. The valuable scientific findings cannot be delivered properly with such an overwhelming linguistic problem. For the same reason it is pointless to list all badly formulated sentences. The authors should rewrite the review after seeking the help of any researcher who is proficient in English.
>> (Response) As your suggestion, professional language editors reviewed and improved the quality of our manuscript's English (Company name: Cambridge Proofreading LLC, Invoice No: 445-12-40). Certificate attached below.
- Paragraphs 4.1 and 4.2: the terms “nanoparticles” and particulate matter should be clearly defined.
>> (Response) Modified defined the term particulate matter and nanoparticles in the beginning of the Paragraphs 4.1 page 9, line 182-183; and Paragraphs 4.2, page 10, line 202-204.

Reviewer 3 Report
The review by Rahman and colleagues is interesting, debating and describing in depth the role of autophagy in eliminating environmental agents. Particularly, this review focuses on the alteration of molecular mechanisms when environmental contaminants damage cells.
Only few comments/corrections:
Title of Paragraph 2 "Recently studies autophagy pathways to target environmental toxic clearance". The title is not clear. If I correctly understood, I would suggest "Autophagy pathways to target environmental toxic clearance". Furthermore, this paragraph includes many acronyms. The first time you use the term, put the acronym in parentheses after the full term. Thereafter, you can stick to use the acronym. So, please provide for this in the paragraph and throughout the entire text of the manuscript. Lastly, please indicate that SQSTM1 and p62 are the same molecule.
In Figure 4 and 5, please replace P62 with p62.
In Paragraph 4.1, the Authors describe the elimination of particulate matter by autophagy. To be thorough, I would consider worthwhile to mention also the manuscript by Colasanti et al. for the description of particulate matter-induced autophagy in human bronchial epithelial cells ("Diesel exhaust particles induce autophagy and citrullination in Normal Human Bronchial Epithelial cells", Colasanti et al., Cell Death Dis. 2018 Oct 19;9(11):1073. doi: 10.1038/s41419-018-1111-y.)
Author Response
The review by Rahman and colleagues is interesting, debating and describing in depth the role of autophagy in eliminating environmental agents. Particularly, this review focuses on the alteration of molecular mechanisms when environmental contaminants damage cells.
>> (Response) First of all, we would like to express our sincere gratitude for the time and effort the reviewer had put into reviewing our manuscript. We have incorporated changes based on the reviewer comments provided in the manuscript which revised parts are highlighted by BLUE color in the entire revised manuscript.
Only few comments/corrections:
Title of Paragraph 2 "Recently studies autophagy pathways to target environmental toxic clearance". The title is not clear. If I correctly understood, I would suggest "Autophagy pathways to target environmental toxic clearance". Furthermore, this paragraph includes many acronyms. The first time you use the term, put the acronym in parentheses after the full term. Thereafter, you can stick to use the acronym. So, please provide for this in the paragraph and throughout the entire text of the manuscript. Lastly, please indicate that SQSTM1 and p62 are the same molecule.
>> (Response) We modified the paragraph accordingly. ‘2. Autophagy pathways to target environmental toxin substances clearance’. Additionally, we indicate SQSTM1 and p62 are the same molecule. Page 5, line 88.
In Figure 4 and 5, please replace P62 with p62.
>> (Response) We modified in figure 4 (page 8) and 5 (page 11).
In Paragraph 4.1, the Authors describe the elimination of particulate matter by autophagy. To be thorough, I would consider worthwhile to mention also the manuscript by Colasanti et al. for the description of particulate matter-induced autophagy in human bronchial epithelial cells ("Diesel exhaust particles induce autophagy and citrullination in Normal Human Bronchial Epithelial cells", Colasanti et al., Cell Death Dis. 2018 Oct 19;9(11):1073. doi: 10.1038/s41419-018-1111-y.)
>> (Response) We added the mentioned article in our manuscript and cite it. Page 10, line 194-198.

Round 2
Reviewer 2 Report
The review by Rahman et al. has been substantially improved from the linguistic point of view and the content is now easier to follow, but still contains many incomprehensible ideas and sentences from the scientific point of view. The review continues to be confusing and many subtitles continue to be badly formulated or do not match the content of chapters that they should describe. The authors have addressed only some issues pointed out in the review 1. In addition, the new revised version exposes several other conceptual and scientific issues that should be addressed before resubmitting this manuscript. The major points of criticism are listed below.
Major points:
1. The title has been improved but there is a grammar error inside. It should be: The emerging role of autophagy as a target of environmental pollutants: an update on mechanisms.
2. The abstract has been shortened and now it is less confusing, but it still fails to deliver a point to point summary of the main topics addressed in the review. For example, part of the review focuses on different mechanisms involved in autophagy of solid dust particles, nanoparticles, metal and metalloids and in smoke-induced autophagy, but the abstract does not mention it. There are too many generic sentences but little information on the purpose and content of the review. The authors should clearly state the purpose and main points that are discussed in the review.
3. The titles of chapters have been improved, but in most cases need to be rephrased as either do not match the content of the chapter and/or are conceptually difficult to follow. Please see comments to each chapter.
4. Chapter 2 title in line 105: “ 2. Autophagy pathways to target environmental toxin substances clearance.”
First problem: Is autophagy targeting the clearance of environmental toxic substances? Which substances? Second problem: The title does not fit the content of chapter 2, which describes different types of cargo-specific autophagy and related receptors. This chapter does not explain which environmental toxins the authors are going to talk about.
5. Chapter 3: The title is still unclear. It is difficult to understand what the authors mean by this title. After reading this chapter it also becomes clear that a different title should be applied. For example: “Effects of pesticides and other small molecular weight environmental toxins on autophagy”. This chapter is extremely chaotic as the authors do not try to organize the scientific facts in those that indicate a dysregulation of autophagy by excessive stimulation and its inhibition. The authors also do not cite and discuss many crucial papers regarding pesticide and arsenic effects on autophagy, for example: Mader et al, 2012, Janda et al. 2012; Wills et al. 2012; Dagda et al. 2013; Giordano et al. 2014; Janda et al. 2015; Liu J, Liu W et al. 2018; Bae J et al, 2018; Cui Y. H., 2021.
6. Chapter 4: The title again is conceptually wrong. The word “application” can be used with respect to methods and drugs, but we cannot talk about “application” of biological processes. These can be stimulated or inhibited like autophagy to protect or eliminate depending on the context. The subtitles of some paragraphs are also conceptually wrong. Are heavy metals PM, and nanoparticles really eliminated by autophagy? Was it clearly demonstrated? Or maybe the biological systems upregulate autophagy as defense mechanism in response to the toxin and pollutants, as they were trying to get rid of these pollutants by enhancing the autophagy flux. The discussion of scientific evidence in chapter 4 should organized in examples positive and negative effects on autophagy. Each chapter should finish with one or two sentence that deliver a clear conclusive message.
7. The paragraph 4.4 on smoke -induced autophagy is particularly chaotic with respect to other paragraphs. First, the title does not correspond to the content. Is the smoke really eliminated by autophagy? The presented evidence does not suggest it. The authors should first explain what is defined as “smoke” and what compounds are commonly found in the cigarette or other types of smoke. Then they should talk about the evidence showing how smoke compounds affect autophagy pathways. This evidence should also be organized in positive and negative effects on autophagy.
8. The entire text of this review needs to be better organized. Sentences are often out of context, inconclusive or just unclear. Below are the examples of badly formulated sentences. The list stops at position 20, but could be at least two times longer.
Minor issues:
1. The sentence ln lines 38-40 is incorrect: “Many environmental pollutants have been found to influence autophagy under stress conditions, particularly in chemical stress, which can increase autophagy.” The second part of the sentence: “under stress conditions ……” should be eliminated, since it is not true that pollutants influence autophagy only under stress conditions, particularly chemical stress …. What is “chemical stress”? Please define or reformulate.
2. Line 55: What is “a cytostatic connection”, maybe the authors mean “a cytostatic mechanism” or simply: “…is cytostatic in cancer cell …”
3. Line 57: “often” is incorrect. It should be: “PAS ….initiate the autophagy cascade.”
4. Line 64: the term “phagocytosis” is reserved for an encapsulation of extracellular cargo. Use another term or other terms. For the same reason the reference 6 is not an appropriate reference.
5. Line 71: The second part of the sentence: “a critical mechanism for cellular repair and survival mechanism.” Is redundant. It maybe used in the first descriptive sentences.
6. Line 72: a reference is needed. The authors may cite Janda et al., Autophagy 2015 or other suitable literature.
7. Figure 1: “mTOR independent” should be specified at least as “mTOR independent pathway.”
8. Line 108: Please rephrase the entire sentence as the word “labels” is not appropriate.
9. Line 111: Please rephrase the entire sentence as the expression “some significant” is not appropriate. Maybe the authors meant “most important”?
10. The ref.s 17 e 18 are not appropriate. Please add a relevant citation, covering different types of autophagy, from top journals in the field.
11. Line 115: Autophagy selectively degrades lipid droplets. Please rephrase or it can be: “ Lipophagy selectively degrades lipid …”
12. Line 115-116: The next sentence regarding SQSTM1/p62 does not make any sense. What does it mean “…to connect through autophagy”??? Please rephrase.
13. The authors should use one name either SQSTM1 or p62, but not both names throughout the Ms.
14. Line 137: Which is this “unifying principle”? Please finish the sentence or explain.
15. Line 141-143: please add one sentence suggesting examples of autophagy receptors being external or integral components of cargoes.
16. Line 145-147: Please rephrase this meaningless sentence. Aggregophagy in yeast???? mediates what ???? in neurodegenerative disorders???.
17. Line 148-150: This is the worst sentence in the review. It is totally out of the context. Please explain what you mean by this sentence. Please rephrase: Environmental toxicities arising from what????
18. Line 158: “The relative cytotoxicities of nanoparticles/nanodrugs and metalloids/metals in cells …compared. This sentence is inconclusive. What is the conclusion after this comparison?
19. Line 159-161: The sentence “While Autophagy’s role in carcinogenesis is …. “ is out of context.
20. Line 162: the simple present tense should be used “act” as it is a common findings.
Author Response
The review by Rahman et al. has been substantially improved from the linguistic point of view and the content is now easier to follow, but still contains many incomprehensible ideas and sentences from the scientific point of view. The review continues to be confusing and many subtitles continue to be badly formulated or do not match the content of chapters that they should describe. The authors have addressed only some issues pointed out in the review 1. In addition, the new revised version exposes several other conceptual and scientific issues that should be addressed before resubmitting this manuscript. The major points of criticism are listed below.
>> (Response) First of all, we would like to express our sincere gratitude for the time and effort the reviewer had put into reviewing our manuscript. We have incorporated changes based on the reviewer comments provided in the manuscript which revised parts are highlighted by BLUE color in the entire revised manuscript.
Major points:
- The title has been improved but there is a grammar error inside. It should be: The emerging role of autophagy as a target of environmental pollutants: an update on mechanisms.
>> (Response) We accepted the reviewer proposed title.
- The abstract has been shortened and now it is less confusing, but it still fails to deliver a point to point summary of the main topics addressed in the review. For example, part of the review focuses on different mechanisms involved in autophagy of solid dust particles, nanoparticles, metal and metalloids and in smoke-induced autophagy, but the abstract does not mention it. There are too many generic sentences but little information on the purpose and content of the review. The authors should clearly state the purpose and main points that are discussed in the review.
>> (Response) We modified the abstract according to the reviewer comments and mared with BLUE color.
- The titles of chapters have been improved, but in most cases need to be rephrased as either do not match the content of the chapter and/or are conceptually difficult to follow. Please see comments to each chapter.
>> (Response) We modified the chapter title accordingly.
- Chapter 2 title in line 105: “ 2. Autophagy pathways to target environmental toxin substances clearance.”
First problem: Is autophagy targeting the clearance of environmental toxic substances? Which substances? Second problem: The title does not fit the content of chapter 2, which describes different types of cargo-specific autophagy and related receptors. This chapter does not explain which environmental toxins the authors are going to talk about.
>> (Response) This is wonderful suggestion from the reviewer. This chapter we explained using several autophagy processes may be targeted to clean environmental toxin substances like metals, airborne particulate matter, nanoparticles, and cigarette smoke. As we describe the process of autophagy pathway, later we detail describe how autophagy pathway might be clean environmental exposure. Finally, we changed the chapter title ‘Mechanism of autophagy pathways.
- Chapter 3: The title is still unclear. It is difficult to understand what the authors mean by this title. After reading this chapter it also becomes clear that a different title should be applied. For example: “Effects of pesticides and other small molecular weight environmental toxins on autophagy”. This chapter is extremely chaotic as the authors do not try to organize the scientific facts in those that indicate a dysregulation of autophagy by excessive stimulation and its inhibition. The authors also do not cite and discuss many crucial papers regarding pesticide and arsenic effects on autophagy, for example: Mader et al, 2012, Janda et al. 2012; Wills et al. 2012; Dagda et al. 2013; Giordano et al. 2014; Janda et al. 2015; Liu J, Liu W et al. 2018; Bae J et al, 2018; Cui Y. H., 2021.
>> (Response) We appreciated reviewer comments and replaced new chapter title with some new references ‘Effects of pesticides and other small molecular weight environmental toxins on autophagy’.
- Chapter 4: The title again is conceptually wrong. The word “application” can be used with respect to methods and drugs, but we cannot talk about “application” of biological processes. These can be stimulated or inhibited like autophagy to protect or eliminate depending on the context. The subtitles of some paragraphs are also conceptually wrong. Are heavy metals PM, and nanoparticles really eliminated by autophagy? Was it clearly demonstrated? Or maybe the biological systems upregulate autophagy as defense mechanism in response to the toxin and pollutants, as they were trying to get rid of these pollutants by enhancing the autophagy flux. The discussion of scientific evidence in chapter 4 should organized in examples positive and negative effects on autophagy. Each chapter should finish with one or two sentence that deliver a clear conclusive message.
>> (Response) We modified chapter 4 with new title: Targeting autophagy modulation to eliminate environmental substances. It has been found that PM has been associated with various health conditions, especially respiratory illnesses which associated PM exposure with autophagy and airway dysfunction. Additionally, another of the mechanisms of intrinsic toxicity that are exhibited by NPs is the disruption of autophagy. The disruption of autophagy that NPs cause must be understood in order to ensure the safety of nanotechnology. Therefore, targeting autophagy is an important factor to eliminate PM and NP from environmental.
- The paragraph 4.4 on smoke -induced autophagy is particularly chaotic with respect to other paragraphs. First, the title does not correspond to the content. Is the smoke really eliminated by autophagy? The presented evidence does not suggest it. The authors should first explain what is defined as “smoke” and what compounds are commonly found in the cigarette or other types of smoke. Then they should talk about the evidence showing how smoke compounds affect autophagy pathways. This evidence should also be organized in positive and negative effects on autophagy.
>> (Response) We significantly modified according to the suggestion and marked as BLUE color in chapter 4.4 Elimination of smoke by autophagy.
- The entire text of this review needs to be better organized. Sentences are often out of context, inconclusive or just unclear. Below are the examples of badly formulated sentences. The list stops at position 20, but could be at least two times longer.
Minor issues:
- The sentence ln lines 38-40 is incorrect: “Many environmental pollutants have been found to influence autophagy under stress conditions, particularly in chemical stress, which can increase autophagy.” The second part of the sentence: “under stress conditions ……” should be eliminated, since it is not true that pollutants influence autophagy only under stress conditions, particularly chemical stress …. What is “chemical stress”? Please define or reformulate.
>> (Response) We modified the sentence according to the reviewer comments.
- Line 55: What is “a cytostatic connection”, maybe the authors mean “a cytostatic mechanism” or simply: “…is cytostatic in cancer cell …”
>> (Response) We corrected.
- Line 57: “often” is incorrect. It should be: “PAS ….initiate the autophagy cascade.”
>> (Response) We modified and corrected.
- Line 64: the term “phagocytosis” is reserved for an encapsulation of extracellular cargo. Use another term or other terms. For the same reason the reference 6 is not an appropriate reference.
>> (Response) As it is not relevant, we deleted the sentence.
- Line 71: The second part of the sentence: “a critical mechanism for cellular repair and survival” Is redundant. It maybe used in the first descriptive sentences.
>> (Response) As it is not relevant, we deleted the sentence.
- Line 72: a reference is needed. The authors may cite Janda et al., Autophagy 2015 or other suitable literature.
>> (Response) We modified and corrected.
- Figure 1: “mTOR independent” should be specified at least as “mTOR independent pathway.”
>> (Response) We added mTOR independent pathway in page 3 line 54-59.
- Line 108: Please rephrase the entire sentence as the word “labels” is not appropriate.
>> (Response) We modified the sentence page 5 line 86-89.
- Line 111: Please rephrase the entire sentence as the expression “some significant” is not appropriate. Maybe the authors meant “most important”?
>> (Response) We deleted and modified.
- The ref.s 17 e 18 are not appropriate. Please add a relevant citation, covering different types of autophagy, from top journals in the field.
>> (Response) Corrected and added appropriate references.
- Line 115: Autophagy selectively degrades lipid droplets. Please rephrase or it can be: “ Lipophagy selectively degrades lipid …”
>> (Response) We modified and connected.
- Line 115-116: The next sentence regarding SQSTM1/p62 does not make any sense. What does it mean “…to connect through autophagy”??? Please rephrase.
>> (Response) We modified and connected.
- The authors should use one name either SQSTM1 or p62, but not both names throughout the Ms. >> (Response) Corrected.
- Line 137: Which is this “unifying principle”? Please finish the sentence or explain.
>> (Response) As it is not relevant, we deleted the sentence.
- Line 141-143: please add one sentence suggesting examples of autophagy receptors being external or integral components of cargoes.
>> (Response) Modified in page 6 line 112-114.
- Line 145-147: Please rephrase this meaningless sentence. Aggregophagy in yeast???? mediates what ???? in neurodegenerative disorders???.
>> (Response) Modified and corrected.
- Line 148-150: This is the worst sentence in the review. It is totally out of the context. Please explain what you mean by this sentence. Please rephrase: Environmental toxicities arising from what????
>> (Response) Modified and replaced by another sentence page 6 line 119-122.
- Line 158: “The relative cytotoxicities of nanoparticles/nanodrugs and metalloids/metals in cells …compared. This sentence is inconclusive. What is the conclusion after this comparison?
>> (Response) As it is not relevant, we deleted the sentence.
- Line 159-161: The sentence “While Autophagy’s role in carcinogenesis is …. “ is out of context.
>> (Response) As it is not relevant, we deleted the sentence.
- Line 162: the simple present tense should be used “act” as it is a common findings.
>> (Response) We modified.